# Usefulness of Complete Blood Count (CBC) to Assess Cardiovascular and Metabolic Diseases in Clinical Settings: A Comprehensive Literature Review

**DOI:** 10.3390/biomedicines10112697

**Published:** 2022-10-25

**Authors:** In-Ho Seo, Yong-Jae Lee

**Affiliations:** Department of Family Medicine, Yonsei University College of Medicine, Seoul 06273, Korea

**Keywords:** complete blood count (CBC), cardiovascular disease (CVD), metabolic diseases, type 2 diabetes (T2DM)

## Abstract

Complete blood count (CBC) is one of the most common blood tests requested by clinicians and evaluates the total numbers and characteristics of cell components in the blood. Recently, many investigations have suggested that the risk of cancer, cardiovascular disease (CVD), arteriosclerosis, type 2 diabetes (T2DM), and metabolic syndrome can be predicted using CBC components. This review introduces that white blood cell (WBC), neutrophil-to-lymphocyte ratio (NLR), hemoglobin (Hb), mean corpuscular volume (MCV), red cell distribution width (RDW), platelet count, mean platelet volume (MPV), and platelet-to-lymphocyte ratio (PLR) are useful markers to predict CVD and metabolic diseases. Furthermore, we would like to support various uses of CBC by organizing pathophysiology that can explain the relationship between CBC components and diseases.

## 1. Introduction

Complete blood count (CBC) is one of the most common blood tests requested by clinicians [1] and evaluates the total numbers and characteristics of cell components in the blood, such as white blood cells (WBCs), red blood cells (RBCs), and platelets. CBC includes: (1) WBC total and differential count; (2) erythrogram (RBC count, determination of hemoglobin (Hb) and hematocrit, and indices calculation (mean corpuscular volume (MCV), mean corpuscular hemoglobin (MCH), mean corpuscular hemoglobin concentration (MCHC), and red cell distribution width (RDW))); and (3) platelet count indices calculation (mean platelet volume (MPV)) [2]. The reference ranges of each component of CBC are shown in Table 1. In addition to the WBC, RBC, and platelet parameters per se, the combination of CBC components such as the neutrophil-to-lymphocyte ratio (NLR), platelet-to-lymphocyte ratio (PLR), and monocyte-to-lymphocyte ratio (MLR) have been investigated in previous epidemiological studies.

CBC results are used to assess acute or chronic infections if WBCs are increased, leukemia when WBCs are increased or decreased, anemia if Hb is low, and liver cirrhosis when platelet counts are decreased [2]. However, previous investigations have suggested that the risk of cancer, cardiovascular disease (CVD), arteriosclerosis, type 2 diabetes (T2DM), and metabolic syndrome can be predicted using the CBC components mentioned above and secondary results in a specific combination of CBC components.

This review introduces several studies that have investigated the relationship between specific CBC components and CVD and metabolic diseases and suggests that CBC can be a useful tool for managing CVD and metabolic diseases.

## 2. WBC Parameters

### 2.1. WBC Count

One of the most important indicators in interpreting CBC results is WBC count. Several types of circulating WBCs can be classified according to differentiation process, size, and shape, including neutrophils, lymphocytes, monocytes, eosinophils, and basophils. WBCs are an essential part of the immune system, and the different types of WBCs have distinct roles in the human body. For example, an increase in WBCs can be a sign of infection in any part of the body and can indicate the presence of a tumor. A decrease in WBCs can suggest a problem with bone marrow production or be a side effect of certain drugs, including anticancer drugs. In addition, the WBC component of CBC is clinically useful in follow-up to determine the therapeutic effects of treatments for infection or inflammation. Therefore, interpretation of the CBC test is important in clinical settings; greater interest in and careful interpretation of abnormalities in WBC and leukocyte differential calculations are needed. Studies on the relationship between WBC count and CVD or metabolic diseases have demonstrated that WBC increase is associated with incidence of disease or disease severity. Because prior studies have shown that WBC count increases in inflammatory situations and CVD and that metabolic diseases are related to chronic inflammation, the relationship between WBC increase and CVD can be interpreted easily and quickly using CBC results.

#### 2.1.1. WBC Count and CVD

In large mortality studies, a WBC count increase has been found to be associated with an increased risk of CVD mortality. The Atherosclerosis Risk in Communities (ARIC) Study investigated an average of 13,555 African–American and White men and women who did not have CVD or cancer over 8 years of follow-up [3].

In the ARIC study, participants were classified into quartiles by baseline WBC count. The risk of CVD mortality in the highest quartile of WBC count (≥7000 cells/mm^3^) was 2.3 times higher (95% confidence interval (CI): 1.58, 3.44; *p* < 0.001) compared with that in the lowest WBC count. Another large population-based cohort study, the UK Biobank study, examined more than 500,000 participants and showed that the highest decile of WBC count had a higher risk of CVD mortality than those in the lowest decile (hazard ratio (HR): 1.64, 95%; CI: 1.24–2.16) [4]. A WBC count increase is associated with a higher incidence of nonfatal CVD as well as increased CVD mortality. In the UK Biobank cohort study, both men and women in the highest decile of WBC count had a higher risk of nonfatal CVD than those in the fifth decile (men: HR: 1.28, 95% CI: 1.16–1.42; women: HR: 1.21, 95% CI: 1.06–1.38) [4]. In addition, this trend was evident in the relationship between an increase in neutrophils and CVD incidence risk.

An increased neutrophil count in the Carbohydrates, Lipids, and Biomarkers of Traditional and Emerging Cardiometabolic Risk Factors (CALIBER) study, a cohort study that included more than 700,000 people, was associated with an increase in 12 CVD incidents [5]. In a study with a median follow-up time of 11 years that assessed the associations of prediagnostic CBC results with the incidence of CVD, a higher total WBC count was associated with a high risk of CVD (HR: 1.31, 95% CI: 1.10–1.55) [6]. These studies show that a WBC increase in the general population can have sufficient potential to predict CVD risk or CVD mortality. In a previous study, WBC count and other subtypes of WBCs, including monocytes, lymphocytes, and eosinophils, were also positively associated with coronary heart disease (CHD), peripheral arterial disease, and stroke [7]. Furthermore, the authors suggested that the CHD risk ratios associated with a high WBC count are comparable to those of other inflammatory markers, including C-reactive protein [7].

#### 2.1.2. WBC Count and Metabolic Diseases

A relationship between WBC and disease risk has been reported in metabolic diseases as well as CVD. A cross-sectional study conducted in China investigated the relationship between total WBC count, arterial stiffness, and T2DM [8]. That study showed an increased incidence of T2DM with increasing brachial–ankle pulse wave velocity (baPWV) and suggested that WBC count affects the pathogenesis of insulin resistance. When participants were distributed into three groups according to WBC level, the incidence of T2DM in the highest WBC group increased compared with that in the lowest group (OR: 1.54, 95% CI: 1.01–2.35). Another longitudinal study with a large number of participants and a long follow-up period showed that the incidence of T2DM was associated significantly and prospectively with WBC count (4th vs. 1st quartile HR: 1.37, 95% CI: 1.22–1.53) after adjustments for potential confounding factors [9].

Metabolic syndrome is associated with increased WBCs. A cross-sectional study that analyzed 203 older adults with an average age of 80.2 years showed that the chance of developing metabolic syndrome was 2.4 times higher than when using the cut-off points of a receiver operating characteristic curve [10]. In a cross-sectional study conducted in a young population whose average age was 41.3 years, an increase in WBCs indicated that the chances of developing metabolic syndrome were high even after calibrating for other confounding factors (OR: 1.26, 95% CI: 1.01–1.58) [11].

### 2.2. Neutrophil-to-Lymphocyte Ratio

The most frequent type of WBC is neutrophils, accounting for 50–70% of the total WBC in blood circulation [12]. Neutrophils are the primary immune cells that respond to infection and are controlled under homeostatic conditions [12]. Lymphocytes are a critical population of WBCs and play an essential role in both innate and adaptive immunity [12]. They constitute 20–50% of total WBCs and consist of T cells, B cells, natural killer T cells, and innate lymphoid cells [12]. The ratio obtained by dividing neutrophils by lymphocytes is called the neutrophil–lymphocyte ratio (NLR) and is a useful biomarker that indicates the balance between systemic inflammation and immunity [13]. NLR has been investigated as a predictive and prognostic marker in several diseases, such as various cancers and CVD [14,15]. One advantage of using NLR is that, even when the WBC count is in the normal range, if NLR is 2.5 or higher, clinicians can identify chronic low-grade systemic inflammation [16]. In addition, NLR is useful in clinical practice because it can be calculated easily from CBC using the differential count result.

#### 2.2.1. NLR and CVD

Many previous studies have shown that NLR is an accurate predictor of several diseases through a reflection of systemic inflammation and immune balance. In a study using 1999–2014 National Health and Nutrition Examination Survey Mortality-linked data, NLR was divided into quartiles to investigate overall mortality [14]. An elevated NLR was associated with overall mortality (HR: 1.14, 95% CI: 1.10–1.17, per quartile NLR) and mortality due to heart disease (HR: 1.17, 95% CI: 1.06–1.29) [14]. In addition, in healthy people without known diseases, NLR had a positive association with mortality from heart disease [14]. The results of this prior study showed that NLR could predict CVD mortality in the general population, which suggests that inflammation and immunity can affect disease progression.

Epidemiological studies showing the relationship between NLR and CVD began to emerge in 2005, and numerous reports have shown the potential of NLR as a predictor of CVD [17]. In a systematic review and meta-analysis that investigated NLR and CVD risk, high NLR was significantly associated with risk of coronary artery disease (CAD), acute coronary syndrome (ACS), stroke, and composite cardiovascular events (CAD OR: 1.62, 95% CI: 1.38–1.91; ACS OR: 1.64, 95% CI: 1.30–2.05; stroke OR: 2.36, 95% CI: 1.44–2.89; CVD events OR: 3.86, 95% CI: 1.73–8.64) [18]. These results suggest that NLR can be a useful indicator of CVD progression. A recent paper explained that this causal pathway acts as a predictor for CVD. T2DM, creatinine, and hypertension indirectly explained 56% of the total effect of NLR, and the remaining 44% was due to a direct effect, suggesting a meditation effect of NLR for CVD [19]. The relationship between NLR and atherosclerotic event was analyzed in the Canakinumab Anti-Inflammatory Thrombosis Outcomes Study (CANTOS), the Justification for the Use of Statins in Primary Prevention: An Intervention Trial Evaluating Rosuvastatin (JUPITER), the Studies of PCSK9 Inhibition and the Reduction of Vascular Events (SPIRE)-1, SPIRE-2, and the Cardiovascular Inflammation Reduction Trial (CIRT) studies, which included 60,087 participants who received a placebo or canakinumab, rosuvastatin, bococizumab, or methotrexate [20]. Those studies revealed that NLR could predict CVD events and all-cause mortality independently. In addition, this randomized trial study showed that anti-inflammation therapy could reduce NLR, indicating NLR as a clinical biomarker that can be used to monitor CVD and metabolic diseases during follow-up.

The relationship between CAD and NLR has been studied extensively. NLR was associated with calcium score and risk factors for CAD, was associated independently with arterial stiffness, and was suggested to be a high predictive marker of arterial stiffness [21,22]. Because endothelial dysfunction is caused by neutrophil–endotherm interactions, an NLR increase could increase the risk of CAD. Interestingly, improvement in arterial stiffness in ST-elevation myocardial infarction (STEMI) patients who successfully underwent percutaneous coronary intervention (PCI) was associated with a decrease in NLR [23]. In addition, NLR has been used to predict the occurrence of major adverse cardiovascular events when patients with CAD undergo noncardiac surgery [24]. These findings indicate NLR as a clinical marker to confirm the improvement of CAD patients and an indicator to prevent and predict the occurrence of CAD.

#### 2.2.2. NLR and Metabolic Diseases

Because T2DM is associated with chronic inflammation, we can presume that NLR is associated with T2DM. In a cross-sectional study, participants were distributed into four groups (normal, prediabetic, newly diagnosed diabetic, and previously diagnosed diabetic without complication groups) [25]. NLR was sequentially higher in the prediabetic, newly diagnosed, and previously diagnosed diabetic without complication groups than in the normal group. In addition, a cross-sectional study that investigated whether diabetic complications are associated with NLR in patients with T2DM showed that high NLR was associated with an increased prevalence of CVD and diabetic kidney disease (DKD) [26]. The relationship between NLR and diabetic retinopathy, one of the most important complications that can occur in T2DM patients, was investigated. Patients with diabetic retinopathy showed higher NLR than those with nondiabetic retinopathy [27]. In addition, NLR was an independent risk factor for diabetic retinopathy. Although more research is needed to confirm the findings of these cross-sectional studies, NLR has sufficient potential to predict T2DM and its related complications.

## 3. RBC Parameters

RBC count is another crucial CBC result. RBCs are created in the bone marrow and released into the peripheral blood circulation after maturation [28]. RBCs are usually uniform in size and shape, but their appearance can be affected by various conditions, such as iron deficiency [28,29]. Anemia is an illness in which the number of RBCs or Hb concentration is lower than the normal reference range. Many components of CBC are associated with RBC count, such as Hb, hematocrit, MCV, MCH, MCHC, and RDW [2]. We describe the relationships of Hb, MCV, RDW, and CVD with metabolic disease.

### 3.1. Hemoglobin (Hb)

Hb has a tetrameric structure of 97% in the form of α2β2 made of α-, β-globins, 2% in the form of α2γ2 made of α-, γ-globins, and up to 2.5% in the form of fetal Hb, which is the primary Hb that is produced by the fetus during pregnancy [30]. The primary function of Hb is to move O_2_ to the tissue of the whole body. However, Hb can interact with various gases as well as O_2_, and it also interacts with carbon dioxide and carbon monoxide [30]. Among the nonoxygen-related functions of Hb, the interaction between nitric oxide (NO) and Hb is drawing attention. NO plays a significant role in maintaining homeostasis by increasing oxygen transport to tissues, causing blood vessel expansion and increased blood flow. NO produced by endothelial NO synthase enzymes largely diffuses into surrounding smooth muscle and regulates vascular tone [31]. However, under the circumstances of increased intracellular Hb concentrations, such as polycythemia vera, intracellular Hb acts as an efficient scavenger of NO, causing vasoconstriction [30]. Considering this role of Hb, we can infer that the relationship between Hb and vascular-related diseases is possibly connected.

#### Hb and CVD

The easiest status to identify using Hb is anemia. The causes of anemia vary, but the most common is iron deficiency [29]. Anemia is common in heart failure (HF) patients; 27.8% were diagnosed with anemia in a cohort study of HF patients aged 73 in the United Kingdom, and 43.2–68.0% were diagnosed with iron-deficiency anemia [32]. A low level of Hb (HR: 0.92, 95% CI: 0.89–0.95, *p* < 0.001) and a low concentration of serum iron (HR: 0.98, 95% CI: 0.97–0.99, *p* = 0.007) were independently associated with higher all-cause and cardiovascular mortality in multivariable analyses [32]. Iron-deficiency anemia is often present in patients with CAD and ACS. A cohort study with an average follow-up of 9.81 years that included patients with CAD and ACS investigated whether markers of immune activation were associated with disease severity of CAD and ACS (neopterin, interleukin (IL)-12, IL-6, high-sensitivity C-reactive protein (hs-CRP), fibrinogen, serum amyloid A and iron metabolism, ferritin, transferrin saturation, Hb) [33]. Interestingly, among these several inflammatory markers, only Hb could predict an unfavorable outcome of CAD and ACS. Anemia of chronic disease was associated with a higher cardiocerebrovascular event rate in the following 2 years compared with patients with other types of anemia or those without anemia (14.3 vs. 6.1 vs. 4.0%, *p* < 0.001). Another study showed that anemia was an independent factor for predicting death in myocardial infarction (MI) patients and that Hb level was significantly associated with high-sensitivity cardiac troponin [34]. Therefore, it has been suggested that Hb can be used as a marker for predicting chronic myocardial injury.

It remains unclear whether correcting anemia can prevent symptoms, progression, and mortality of cardiovascular disease in CVD patients. A recent study suggested that intravenous ferric carboxymaltose injections in HF patients improved self-reported patient symptoms of exercise capacity (which is measured using the 6-minute walk test distance in anemic patients) and lower rehospitalization rates [35]. However, anemia treatment did not reduce the mortality of HF patients. Although correction of Hb was effective only in alleviating symptoms, and the experiments were conducted only in HF patients, the study showed that anemia (low Hb) was associated with increased CVD severity and could play a role in disease modification.

### 3.2. RDW and MCV

MCV means the average volume of an RBC. MCV is acquired by multiplying blood volume by hematocrit and dividing that product by the RBC count in that same volume [36]. RDW is calculated by dividing the standard deviation of MCV by MCV and multiplying the result by 100 to yield a percentage value to indicate RBC size heterogeneity. The normal reference range for the RDW is 11–15% [37].

RDW is used to distinguish between different types of anemia. In addition to anemia, the following factors such as age, sex, genetic factors, general function, and dyslipidemia could affect RDW [38].

#### 3.2.1. RDW, MCV, and CVD

A recently published review paper on the relationship between RDW and CVD found that RDW was related to the incidence and disease severity of HF, MI, conventional atherosclerosis, arterial fibrillation, and primary hypertension [39]. Accordingly, that prior study suggested RDW as a new predictive marker that can act as an independent risk factor. RDW is a significant prognostic indicator in both stable vascular diseases and acute conditions such as STEMI [40]. In addition, RDW per se or in combination with other cardiac biomarkers could improve the diagnosis and progression of ACS, heart failure, peripheral artery disease, atrial fibrillation, and ischemic cerebrovascular disease [41]. MCV is used to obtain RDW, but studies on the relationship between MCV alone and CVD are rare. In a recent large-scale population-based study, which included a 9-year mean follow-up duration, participants were distributed into three groups based on MCV level to examine the relationship between MCV level and CVD mortality [42]. Compared with the group with an MCV level of 95 or less, the risk of CVD-related death increased in the group with an MCV of 99 or higher (HR: 1.42, 95% CI: 1.15–1.76). To summarize, we suggest that both RDW and MCV are sufficiently available clinical markers to predict CVD occurrence and its prognosis from CBC results.

#### 3.2.2. RDW, MCV, and Metabolic Diseases

Due to the use of the hemoglobin A1c (HbA1c) to monitor the diagnosis and treatment of T2DM, markers related to RBCs are associated with and can be used to diagnose T2DM and to estimate the progression of the disease using CBC in clinical settings. Based on several studies, RDW is a marker of inflammation and prognosis in T2DM patients [43,44]. Subharshree et al. initially examined the relationship between T2DM and RDW [45]. Their study showed that RDW was associated with brain natriuretic peptide in DM patients with HF. Another study investigated whether RDW was associated with a significantly increased risk of new-onset T2DM [44]. In addition, RDW was positively associated with HbA1c; when RDW increased by one standard deviation, HbA1c increased by 0.10%. Malandrino et al. [46] showed a relationship between RDW and T2DM complications in a population of 2,497 people with T2DM. They concluded that a higher RDW level was correlated with an increased chance of developing vascular complications, HF, MI, stroke, and nephropathy. Previous studies suggest that RDW can be carefully considered for predicting the onset and complications of T2DM with HbA1c and total glucose levels.

## 4. Platelet Parameters

Platelets are anucleated cells with a short lifespan of 7–10 days and are eliminated in the spleen or liver after circulating in the body [47]. The primary function of the platelet is hemostasis. Platelets stick to damaged blood vessel walls or clump together, causing blood clotting and stopping blood loss [48]. In addition, platelets play an essential role in systemic inflammation, and an activated platelet contributes to vascular inflammation and damage, atherogenesis, and thrombosis [49]. Several components related to platelets in CBC are platelet count, MPV, and platelet distribution width (PDW), and each factor reflects the condition of various diseases.

### 4.1. Platelet Count and MPV

Platelet count is a simple indicator of hemostasis in clinical practice. In addition to hemostasis, platelet could be a valuable marker for several diseases. In particular, if the platelet count is higher than 450,000/μL, it is called thrombocytosis, which increases the risk of thrombotic complications [50]. MPV is used to measure platelet size, and MPV represents platelet activity. MPV has an inverse relationship with platelet count. This inverse relationship is related to the maintenance of hemostasis and the preservation of platelet mass [51]. Platelet counts and MPV are affected by various factors such as race, age, gender, smoking habits, alcohol consumption, and physical activity [52].

#### 4.1.1. Platelet Count, MPV, and CVD

Since thrombocytosis forms platelet aggregation and arteriolar microthrombi formation against vascular damage, it is closely related to CVD [50]. Previous studies found that antiplatelet therapy is used because the platelet plays a role in the pathophysiology of STEMI [53,54]. In addition, platelet size is a risk factor showing poor clinical outcomes in ACS [55]. When analyzing the Thrombolysis In Myocardial Infarction Trials database in STEMI patients, a higher platelet count was independently associated with adverse clinical outcomes [56]. Compared with the reference group of platelet counts <200,000/μL, the odds ratio (OR) of adverse clinical outcomes was 1.71 times higher (95% CI: 1.16–2.51, *p* < 0.005) in the group of platelet counts >400,000/μL. Interestingly, platelet counts have also been associated with total and cardiovascular death in apparently healthy men [57].

Slacka et al. showed that an increase in MPV was another independent risk factor in patients after acute ischemic cardiac events [58]. Actually, individuals taking clopidogrel were found to be protected from an increase in MPV after acute MI attacks [58].

Moreover, MPV was positively and independently associated with subclinical white matter hypersensitivities of presumed vascular origin in middle-aged and older adults without a history of CVD, cancer, or stroke [59]. Mean MPV was significantly higher in the leukoaraiosis group than in the control group (8.4 ± 0.8 and 8.1 ± 1.0, respectively, *p* = 0.036). D’Erasmo et al. also reported higher mortality among patients with significantly increased MPV in patients after cerebral stroke [60].

#### 4.1.2. Platelet Count, MPV, and Metabolic Diseases

In addition to thrombocytosis or thrombocytopenia, high levels of platelet counts, even within normal range, are positively associated with the incident risk of T2DM from longitudinal studies. In a cohort study with a mean follow-up of 8.4 years, the HR of diabetes incidence in the third tertile was higher at 1.28 (95% CI 1.04–1.57) compared with the reference first tertile [61]. In particular, these positive relationships between platelet count and incident T2DM were more prominent in participants with impaired glucose tolerance (HR: 1.45, 95% CI: 1.05–2.00).

Park et al. showed that platelet count was higher in individuals with metabolic syndrome but only in women [52], suggesting there may be sex differences in the relationship between platelet count and metabolic syndrome.

High MPV was positively associated with the presence of metabolic syndrome in patients with T2DM. Compared with the lowest MPV tertile, the OR for metabolic syndrome in the highest tertile was 1.724 (95% CI: 1.199–2.479) after adjusting for confounding variables [62]. In a meta-analysis from 39 case-control and cross-sectional studies, Zaccardi et al. reported that T2DM subjects tended to have higher MPV and platelet distribution width values compared with individuals without T2DM (standardized mean difference 0.70, 95% CI: 0.50–0.91, n = 24,245) [63].

### 4.2. Platelet and Lymphocyte Ratio (PLR)

Along with NLR, emerging evidence suggests that PLR has also been proposed as a marker for various diseases, such as CVD and kidney diseases [25,64]. An elevated PLR level was useful in predicting high-risk heart scores in patients admitted with non-ST-elevation myocardial infarction (NSTEMI) and end-stage renal disease [65].

#### 4.2.1. PLR and CVD

Wang et al. reported that PLR could predict major adverse cardiac events (MACE) at long-term follow-up in patients with STEMI undergoing percutaneous coronary intervention (PCI) [66]. PLR was significantly high in the MACE group (147.62 in the MACE group and 111.19 in the non-MACE group). In patients hospitalized with non-ST-elevation myocardial infarction (NSTEMI), PLR was positively correlated with the HEART score [65].

In addition, a previous study showed that PLR could be a useful biomarker to support the diagnosis of acute lower extremity deep vein thrombosis (DVT) [67].

These studies suggest that the application of PLR to patients with CVD can also be a tool for screening high-risk patients relatively easily and quickly.

#### 4.2.2. PLR and Metabolic Diseases

Several observational studies reported that high PLR is positively associated with people with increased insulin resistance and T2DM [68,69]. However, the relationship between PLR and T2DM differs according to the stage of diabetes. Although PLR was reduced in prediabetes and early diabetes, PLR was increased in the later stage of diabetes [25]. Compared with the PLR of the normal tolerance (NGT) group, impaired glucose tolerance (IGT) group, and newly diagnosed diabetes group, PLR was lower in the IGT and newly diagnosed group than in the NGT group. On the contrary, PLR in individuals previously diagnosed with T2DM was higher compared with the NGT group.

## 5. Pathophysiology

Chronic inflammation is one explanation for the association of WBC count and NLR with CVD-related mortality, CVD incidence prediction, and disease severity. Arterial stiffness or atherosclerosis is closely related to CVD incidence, and an increase in WBC count and NLR is involved in the formation of arterial stiffness and atherosclerosis through interactions with the endothelium and platelets and overactivity of neutrophil extracellular traps [4,21,22]. In addition, a WBC increase causes a decrease in blood flow, especially in cardiac tissue, which can lead to coronary heart diseases and ischemic heart problems [70].

NLR is also thought to be useful as an excellent marker of CVD. An NLR increase indicates a reduction in lymphocytes induced by increased programmed cell death or infiltration of lymphocytes into cardiac tissue, which are both common in CVD [70].

The relationships of T2DM with WBC count and NLR can be explained by the pathophysiology of chronic inflammation. Various metabolic stimuli lead to an increase in monocytes in the WBC subpopulation of peripheral blood. These monocytes differentiate into macrophages and create an inflammatory state [71].

The possible mechanisms by which an inflammatory state lead to T2DM are a disturbance in insulin signaling in the liver by inflammatory molecules such as IL-6, a proinflammatory effect on insulin, or insulin resistance [71].

Several mechanisms have revealed how Hb, RDW, and MCV can play a functional role in CVD or T2DM. The relationship between Hb and CVD can be explained by inflammation. A post hoc analysis of the CANTOS trial showed that inflammation could contribute to anemia [33]. An analysis of MI patients revealed that elevated hs-CRP and anemia were closely related, and Hb level increased during canakinumab treatment, which typically reduces IL-1β-hsCRP and IL-6. That is, if iron homeostasis is disturbed by inflammation, CVD also can be affected [33]. Inducible nitric oxide synthase (iNOS) is related to inflammation and is increased in several diseases [72]. In particular, increased iNOS expression and activity have been documented in heart failure patients with preserved ejection fraction, and cardiac relaxation was improved when iNOS was reduced [73]. Moreover, since Hb induces an increase in iNOS, an increase in intracellular Hb could affect inflammation and CVD [74].

The relationship between RDW and CVD can be explained by the presence of inflammation and oxidative stress [39,41]. Because inflammation activates the renin–angiotensin system, it changes the RBC maturation system, increasing RDW [39]. In addition, oxidative stress can increase RDW by damaging the RBC membrane and affecting bone marrow production of RBCs [39].

An increase in MCV can explain the relationship between CVD and a mechanism that increases blood viscosity and reduces cardiac blood flow [42]. Hematocrit is included when calculating MCV and influences blood viscosity, with a logarithmic relationship between blood viscosity and hematocrit.

The relationship between platelets and various diseases is described by systemic inflammation [75]. When the platelet is activated, various proinflammatory soluble factors are secreted. Platelet factor 4 (PF4, also called CXCL4) is the most abundantly secreted by activated platelets and accumulates in the endothelium [76]. Moreover, stimulated platelets secrete and store IL-1b, P-selectin, and soluble CD40L, which are associated with the inflammatory response [77,78,79]. The action of these platelets contributes to systemic inflammation, vasculature inflammation, atherogenesis, and thrombosis, which could participate in the pathophysiology of various diseases.

## 6. Conclusions

The incidence of CVD and metabolic diseases, including T2DM, is increasing worldwide; it is crucial to diagnose, predict, and prevent these conditions as early as possible [80]. This review suggests that CBC can be an important and valuable marker in patients with CVD and other metabolic diseases. Furthermore, we organized the pathophysiology that can explain the relationship between CBC components and diseases (Figure 1). Clinicians frequently encounter patients with CVD and metabolic diseases in clinical practice. However, the most optimal tests for each condition are not available to all patients, and clinicians who have limited facilities cannot conduct all preferred tests. The most straightforward and informative test that can be conducted clinically is CBC. Chronic inflammation contributes to the development and exacerbation of CVD and other metabolic diseases. In addition, tissue or cell damage caused by changes in hemodynamics contributes to disease progression. CBC offers a variety of information by reflecting the pathogenesis of these diseases. WBC count and NLR, in particular, can reflect systemic inflammation, while RDW and MCV can provide information on inflammation and overall hemodynamic state. Platelets can reflect vascular damage and systemic inflammation and provide disease severity and gender differences. These markers can be obtained directly from CBC results or can be obtained through simple calculations. Furthermore, CBCs are cost-effective, and they are routinely performed in most clinical settings [2]. Interestingly, Anderson et al. reported the usefulness of a CBC-derived risk score in predicting mortality risk in patients with suspected CVD using WBC count, platelet count, hematocrit, Hb, RDW, MCV, and MCHC [81]. The authors proposed the prediction models of various diseases using CBC components. In conclusion, the use of CBC components including WBC, NLR, RDW, and MCV could improve the sensitivity and specificity of detecting and predicting CVD and metabolic diseases as early as possible.

## Figures and Tables

**Figure 1 biomedicines-10-02697-f001:**
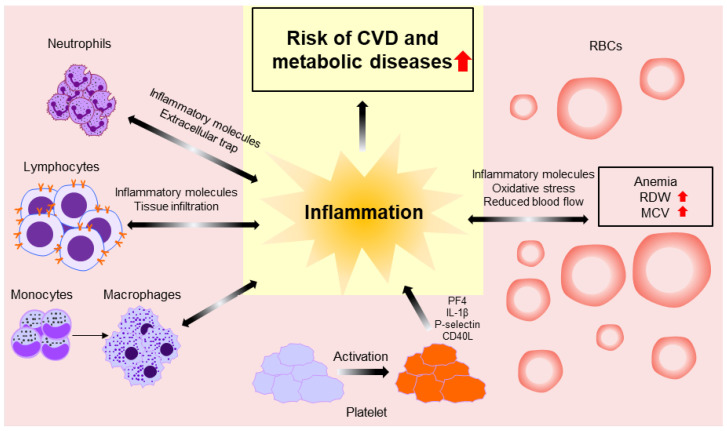
Schematic diagram showing the pathophysiology between CBC components and diseases.

**Table 1 biomedicines-10-02697-t001:** Complete blood count with white blood cell differential count.

Test Acronym	Normal Range Values (Male)	Normal Range Values (Female)
WBC	4.0–10.8 × 10^3^/μL	4.0–10.8 × 10^3^/μL
RBC	4.5–6.1 × 10^6^/μL	4.0–5.4 × 10^6^/μL
Hb	13.0–17.0 g/dL	12.0–16.0 g/dL
HCT	40.0–52.0%	37.0–47.0%
MCV	80–98 fL	80–98 fL
MCH	27.0–33.0 pg	27.0–33.0 pg
MCHC	31.5–37.0 g/dL	31.5–37.0 g/dL
RDW	11.5–14.5%	11.5–14.5%
PLT	150–400 × 10^3^/μL	150–400 × 10^3^/μL
Neutrophils %	40–73%	
Lymphocytes %	19–48%	
Monocytes %	0.4–10.0%	
Eosinophils %	0–7.0%	
Basophils %	0–2.0%	

## Data Availability

Not applicable.

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
