# Peer review of "Usefulness of Complete Blood Count (CBC) to Assess Cardiovascular and Metabolic Diseases in Clinical Settings: A Comprehensive Literature Review"

_biomedicines, 2022, doi:10.3390/biomedicines10112697_

Round 1

Reviewer 1 Report

This is an interesting Review which introduces several studies that have investigated whether specific CBC may be useful tools to predict cardiovascular and metabolic diseases.

The manuscript is clearly written and well organized. Overall, the references are appropriate. I have two suggestions:

(1)The role of HbNO as a reflection of endothelial activity should be more discussed, and include the contribution of inflammation-induced iNOS in some metabolic and inflammatory cardiovascular diseases (such as HFpEF) (PMID 30971818). 

(2)More recent studies should be cited by the authors regarding the relationship between platelet indices (platelet count or MPV) and CVD (PMID 27776204 ; 33939298) and metabolic diseases (PMID 25421610 ; 33787620).

Author Response

Reply to the Reviewer #1

This is an interesting Review which introduces several studies that have investigated whether specific CBC may be useful tools to predict cardiovascular and metabolic diseases. The manuscript is clearly written and well organized. Overall, the references are appropriate. I have two suggestions:

  1. The role of HbNO as a reflection of endothelial activity should be more discussed, and include the contribution of inflammation-induced iNOS in some metabolic and inflammatory cardiovascular diseases (such as HFpEF) (PMID 30971818).

Response: As recommended by the reviewer, we added the role of HbNO in [3.1. Hemoglobin] section and included a paragraph for iNOS in [5. Pathophysiology] section. (page 6, line 211-215, and page 10, line 407-412)

“NO produced by endothelial NO synthase enzymes largely diffuses into surrounding smooth muscle and regulates vascular tone. However, under the circumstances of increased intracellular Hb concentrations such as polycythemia vera, intracellular Hb acts as an efficient scavenger of NO, causing vasoconstriction.”

“Inducible nitric oxide synthase (iNOS) is related to inflammation and increased in cardiovascular and metabolic diseases. In particular, increased iNOS expression and activity have been documented in heart failure patients with preserved ejection fraction and cardiac relaxation was improved when iNOS was reduced. Moreover, since Hb induces an increase in iNOS, an increase in intracellular Hb could affect inflammation and CVD.”

  1. More recent studies should be cited by the authors regarding the relationship between platelet indices (platelet count or MPV) and CVD (PMID 27776204; 33939298) and metabolic diseases (PMID 25421610; 33787620).

Response: Following the reviewer’s suggestion, we added the following sentences in the revised manuscript [4.1.2. Platelet count, MPV and metabolic diseases] section. (page 8, line 346-352)

“High MPV was positively associated with the presence of metabolic syndrome in patients with T2DM. Compared with the lowest MPV tertile, the OR for metabolic syndrome in the highest tertile was 1.724 (95% CI: 1.199–2.479) after adjusting for confounding variables. In a meta-analysis from 39 case-control and cross-sectional studies, Zaccardi et al. reported that T2DM subjects tend to have higher MPV and platelet distribution width values, as compared with individuals without T2DM (standardized mean difference 0.70, 95% CI: 0.50–0.91, n=24,245).”

Reviewer 2 Report

Seo et al proposed to review the use of the Complete Blood cells Count to assess cardiovascular and metabolic diseases in clinical settings. Although the authors reviewed many articles, some important ones were not evaluated and should be included (Vellioglu et al, 201; Parizadeh et al, 2019; Madjid et al, 2013; Uyarel et al, 2012; Anderson et al, 2007).

The concept of CBC need to be reviewed!

CBC includes: 1. Erythrogram (RBC count, determination of hemoglobin and hematocrit, and indices calculation; 2. WBC total and differential count and 3. Platelet count  indices calculation.

Comments:

1. please, review the terms count, indices, CBC markers (line 358) in all manuscript.

2. Line 25 - Delete "Generally"

3. Line 35 - Review information that leukemia is associated with high WBC (not only!)

4. Line 109 - Review the phrase (For example, the most frequent type of leukocyte in the circulation is the neutrophil)

5. Line 168 and others - Please change "patients were divided" by "distributed"

6. Line 185 - Review the statement "A common condition that affects RBC is anemia!!! Anemia is not a disease, but a sign of illness (such as fever!)

7.  Line 191 - If detailing all types of hb, please also include Fetal Hb

8. Line 197- Review this information"NO is transported through the combination of Hb"

9. Line 213 - Please change "Anemia" to "Hb"

10. Line 270 - Review platelet description. It is an anucleated cell!!!

11. Line 308 - Platelet is countable!

12. Line 325 and 326- Please, rewrite the sentence

Finally, the conclusions in lines 280, 329 and 364 are not supported by the cited references and need to be rewritten.

Author Response

Reply to the Reviewer #2

  1. Seo et al proposed to review the use of the complete blood cells count to assess cardiovascular and metabolic diseases in clinical settings. Although the authors reviewed many articles, some important ones were not evaluated and should be included (Vellioglu et al, 201; Parizadeh et al, 2019; Madjid et al, 2013; Uyarel et al, 2012; Anderson et al, 2007).

Response: In compliance with the reviewer’s suggestions, we have added the following sentences. (page 3, line 90-94, and page 7, line 264-268, and page 9, line 366-367, and page 10-11, line 449-453)

[2.1.1. WBC count and CVD] section: Madjid et al., 2013

“In a previous study, WBC count and other subtypes of WBC, including monocytes, lymphocytes, and eosinophils were also positively associated with coronary heart disease (CHD), peripheral arterial disease, and stroke. Furthermore, the authors suggested that the CHD risk ratios associated with a high WBC count are comparable to those of other inflammatory markers including C-reactive protein.

[3.2.1. RDW, MCV, and CVD] section: Uyarel et al., 2012; Parizadeh et al., 2019

“RDW is a significant prognostic indicator in both stable vascular diseases and acute conditions such as STEMI. In addition, RDW per se or in combination with other cardiac biomarkers could improve the diagnosis and progression of ACS, heart failure, peripheral artery disease, atrial fibrillation, and ischemic cerebrovascular disease.”

[4.2.1. PLR and CVD] section: Velioğlu et al., 2019

“Also, a previous study showed that PLR could be useful biomarkers to support the diagnosis of acute lower extremity deep vein thrombosis (DVT).”

[6. Conclusion] section: Anderson et al., 2007

“Interestingly, Anderson et al. reported the usefulness of a CBC derived risk score to predict mortality risk in patients with suspected CVD, using WBC count, platelet count, hematocrit, Hb, RDW, MCV, and MCHC. The authors proposed the prediction models of various diseases using CBC components.”

  1. The concept of CBC need to be reviewed!

CBC includes: 1. WBC total and differential count, 2. Erythrogram (RBC count, determination of hemoglobin and hematocrit, and indices calculation, and 3. Platelet count indices calculation.

Response: We revised the concept of CBC as recommended by the reviewer. (page 1, line 25-29)

“1. WBC total and differential count, 2. Erythrogram (RBC count, determination of hemoglobin (Hb) and hematocrit, and indices calculation (mean corpuscular volume (MCV), mean corpuscular hemoglobin (MCH), mean corpuscular hemoglobin concentration (MCHC), and red cell distribution width (RDW))), and 3. Platelet count indices calculation (mean platelet volume (MPV))”

  1. Please, review the terms count, indices, CBC markers (line 358) in all manuscript.

Response: As suggested by the reviewer, we changed “indices” to “parameters” and “CBC markers” to “CBC components” (page 1, line 16-17, and page 2, line 46, page 5, line 192, and page 7, line 294, and page 1, line 436, and page 11 line 457)

  1. Line 25 - Delete “Generally”

Response: As recommended by the reviewer, we deleted “generally” in the revised manuscript. (page 1, line 25)

“WBC total and differential count, 2. Erythrogram (RBC count, determination of hemoglobin (Hb) and hematocrit, and indices calculation (mean corpuscular volume (MCV), mean corpuscular hemoglobin (MCH), mean corpuscular hemoglobin concentration (MCHC), and red cell distribution width (RDW))), and 3. Platelet count indices calculation (mean platelet volume (MPV))”

  1. Line 35 - Review information that leukemia is associated with high WBC (not only!)

Response: Thank you for your careful corrections and we corrected the mistakes in the revised manuscript as follows. (page 1, line 34-36)

“CBC results are used to assess acute or chronic infections if WBCs are increased, leukemia when WBCs are increased or decreased, anemia if Hb is low, and liver cirrhosis when platelet counts are decreased.”

  1. Line 109 - Review the phrase (For example, the most frequent type of leukocyte in the circulation is the neutrophil)

Response: As recommended by the reviewer, we revised the phrase as follows. (page 4, line 117-118)

“The most frequent type of WBC is neutrophils, accounting for 50-70% of the total WBC in blood circulation.”

  1. Line 168 and others - Please change “patients were divided” by “distributed”

Response: We changed “divided” to “distributed” at following locations in the revised manuscript. (page 3, line 102, and page 5, line 178, and page 7, line 270)

  1. Line 185 - Review the statement “A common condition that affects RBC is anemia!!! Anemia is not a disease, but a sign of illness (such as fever!)

Response: In compliance with the reviewer’s suggestion, we revised the sentence as follow. (page 5, line 196-197)

“Anemia is an illness in which the number of RBC or Hb concentration is lower than normal reference range.”

  1. Line 191 - If detailing all types of Hb, please also include Fetal Hb

Response: In accordance with the reviewer’s comment, we added the sentence in the revised manuscript as follows. (page 5, line 202-204)

“Hb has a tetrameric structure of 97% in the form of α2β2 made of α-, β-globins, 2% in the form of α2γ2 made of α-, γ-globins, and up to 2.5 % in the form of fetal Hb, which is the primary hemoglobin that is produced by the fetus during pregnancy.”

  1. Line 197- Review this information “NO is transported through the combination of Hb”

Response: We agree with the reviewer’s comment that the expression is somewhat vague. To present the information clearly, we deleted that sentence and added new sentences for NO and Hb in the revised manuscript. (page 6, line 211-215)

“NO produced by endothelial NO synthase enzymes largely diffuses into surrounding smooth muscle and regulates vascular tone. However, under the circumstances of increased intracellular Hb concentrations such as polycythemia vera, intracellular Hb acts as an efficient scavenger of NO, causing vasoconstriction.”

*Deleted sentence

“NO is transported through the combination of Hb and NO to play an important role in arterial microcirculation.”

  1. Line 213 - Please change "Anemia" to "Hb"

Response: Thank you for your careful corrections. We changed “anemia” to “Hb” at following location in the revised manuscript. (page 6, line 230-232)

“Interestingly, among these several inflammatory markers, only Hb could predict an unfavorable outcome of CAD and ACS.”

  1. Line 270 - Review platelet description. It is an anucleated cell!!!

Response: In accordance with the reviewer’s comment, we revised the sentence as follow. (page 7, line 295-296)

“Platelets are anucleated cells, with a short lifespan of 7-10 days, are eliminated in the spleen or liver after circulating in the body.”

  1. Line 308 - Platelet is countable!

Response: Thank you for your careful corrections. We corrected the sentence as follow. (page 8, line 336-338)

“In addition to thrombocytosis or thrombocytopenia, high level platelet count even within normal range are positively associated with the incident risk of T2DM from longitudinal studies.”

  1. Line 325 and 326- Please, rewrite the sentence

Response: As recommended by the reviewer, we revised the paragraph as follows. (page 9, line 361-364)

“Wang et al. reported that PLR could predict major adverse cardiac events (MACE) at long-term follow-up in patients with STEMI undergoing percutaneous coronary intervention (PCI). PLR was significantly high in the MACE group (147.62 in the MACE group and 111.19 in the non-MACE group).”

  1. Finally, the conclusions in lines 280, 329 and 364 are not supported by the cited references and need to be rewritten.

Response: In accordance with the reviewer’s comment, we revised sentences and added relevant references as follows. (page 7, line 290-292, and page 9, line 368-369, and page 10, line 407-412)

“Previous studies suggest that RDW can be carefully considered for predicting the onset and complications of T2DM with HbA1c and total glucose levels.”

“These studies suggest that the application of PLR to patients with CVD can also be a tool for screening high-risk patients relatively easily and quickly.”

“Inducible nitric oxide synthase (iNOS) is related to inflammation and increased in cardiovascular and metabolic diseases. In particular, increased iNOS expression and activity have been documented in heart failure patients with preserved ejection fraction and cardiac relaxation was improved when iNOS was reduced. Moreover, since Hb induces an increase in iNOS, an increase in intracellular Hb could affect inflammation and CVD.”
